# HYBRID TRAINING FOR VISION-LANGUAGE-ACTION MODELS

**Pietro Mazzaglia**
Qualcomm AI Research[*]
pmazzagl@qti.qualcomm.com

**Cansu Sancaktar**[†]
University of Tübingen, Max Planck Institute for Intelligent Systems
cansu.sancaktar@tue.mpg.de

**Markus Peschl**
Qualcomm AI Research
mpeschl@qti.qualcomm.com

**Daniel Dijkman**
Qualcomm AI Research
ddijkman@qti.qualcomm.com

## ABSTRACT

Using Large Language Models to produce intermediate thoughts, a.k.a. Chain-of-thought (CoT), before providing an answer has been a successful recipe for solving complex language tasks. In robotics, similar embodied CoT strategies, generating thoughts before actions, have also been shown to lead to improved performance when using Vision-Language-Action models (VLAs). As these techniques increase the length of the model's generated outputs to include the thoughts, the inference time is negatively affected. Delaying an agent's actions in real-world executions, as in robotic manipulation settings, strongly affects the usability of a method, as tasks require long sequences of actions. However, is the generation of long chains-of-thought a strong prerequisite for achieving performance improvements? In this work, we explore the idea of Hybrid Training (HyT), a framework that enables VLAs to learn from thoughts and benefit from the associated performance gains, while enabling the possibility to leave out CoT generation during inference. Furthermore, by learning to conditionally predict a diverse set of outputs, HyT supports flexibility at inference time, enabling the model to either predict actions directly, generate thoughts or follow instructions. We evaluate the proposed method in a series of simulated benchmarks and real-world experiments.

## 1 INTRODUCTION

Despite recent advances in robotics, truly generalist robot policies have long been elusive. Thanks to the joint efforts of collecting large-scale robot data (O'Neill et al., 2024) and making large Vision Language Models (VLM) open-source (Steiner et al., 2024; Tong et al., 2024), we have entered a new era in robotics foundation models. By fine-tuning VLMs on robotic datasets containing actions, we obtain so-called **Vision-Language-Action models (VLAs)** (Kim et al., 2024; Brohan et al., 2023b;a): large policy models that are trained end-to-end to take language instructions and raw camera images as inputs, and output low-level robotic actions.

VLAs possess several advantages over previous work, such as multimodal prompting of the agent and the availability of knowledge from the base pre-trained VLM. However, generalization to out-of-distribution (OOD) settings, e.g., task configurations not available in the robotics fine-tuning dataset, remains challenging. Indeed, the knowledge of the agent is vast

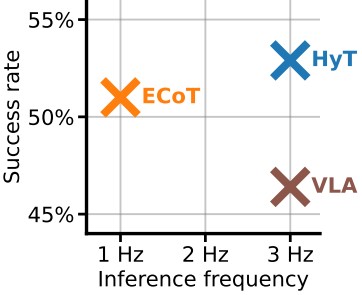

Figure 1: **Hybrid Training (HyT)** of VLAs increases the agent's performance similarly to **ECoT**, but also maintains the same fast inference as standard **VLAs**. Performance refers to the ClevrSkills experiments (9 tasks, 3000 demos) in the Experiments section.

---

[*]Qualcomm AI Research is an initiative of Qualcomm Technologies, Inc.
[†]Work done while doing an internship at Qualcomm AI Research.

about general concepts, but remains limited in the robotics settings, where the data distribution is often narrow.

In order to further unleash the capabilities of VLAs with little data, recent works have trained models to predict intermediate outputs, representing the agent's intentions, before predicting actions (Zawalski et al., 2024; Zhao et al., 2025). One notable example is **Embodied CoT (ECoT)** (Zawalski et al., 2024), where the VLA learns to output useful information about the given task in language form (Wei et al., 2023), before generating the actions to execute. This not only has shown to improve performance, but it also allows humans to more easily interpret the agent's intentions and potentially intervene on them, i.e. correcting the agent's intentions, before action generation. However, due to the intermediate reasoning outputs generated before actions, the action inference frequency of these models can be significantly lower.

The human cognition process from observation to action is hypothesized to leverage the interaction of two systems (Kahneman, 2011). The fast and intuitive **System I** handles most daily tasks, taking control in contexts that our brain judges as unchallenging. The slow and deliberate **System II** is activated when decisions require additional computation, such as comparing options or processing complex information. The tendency of the brain is to delegate as many decisions as possible to System I, to save energy and time. However, in order to do so, humans need to improve their capabilities to deal with complex decisions. This is done by developing a **skilled intuition** (Kahneman & Klein, 2009; Simon, 1992) that allows leveraging previously learned cues to solve familiar tasks, effortlessly.

In this work, we explore the hypothesis that VLA models can similarly develop skilled intuition, when trained with the right objective. Learning from CoT reasoning traces, a model can further internalize knowledge about environments and tasks. Then, at test-time, the model more eagerly recognizes patterns and can leverage such knowledge to generate actions, even in the absence of intermediate thoughts generation. With this hypothesis in mind, we develop a **Hybrid Training (HyT)** framework, which allows the agent to learn from a combination of CoT and actions data.

Hybrid training presents a more flexible learning objective, which encompasses both ECoT and standard VLAs. During training, we implement the HyT objective using a Monte Carlo estimate, consisting of sampling a variety of conditional inputs and outputs with different probabilities. The model learns to predict a set of ouputs, modelling a multitude of conditional action probabilities, which mainly depends on a newly introduced *modality* variable. During test-time, it's the modality variable that allows us to influence the model's generation. By default, the modality variable conditions the model to directly predict actions. This allows VLAs trained with HyT to maintain the same inference time as standard VLAs, while benefitting from training on reasoning traces.

Furthermore, the modality variable can be used for manipulating the VLA into operating in different inference modes. The "act" mode, as mentioned, resembles standard VLA's inference and allows to generate actions directly. In addition, we show a "think" mode, where the VLAs generates intermediate thoughts as in ECoT, and a "follow" mode, where the VLA follows a set of provided intentions, e.g. by a human or an oracle, similarly to the lower level policies in hierarchical systems (Shi et al., 2025; Hafner et al., 2022).

In addition to investigating and validating our proposed HyT framework, on a set of simulated benchmarks (ClevrSkills (Haresh et al., 2024), LIBERO (Liu et al., 2023)) and real-world tasks (on a UFactory xArm 6), we aim to address a fundamental question regarding VLA models:

*What is the contribution of CoT techniques to VLAs performance?*

Our results and analysis support the idea that one of the main utilities of CoT-based training is improving the representation learned by the model (Chen et al., 2025), enabling improved performance, even when no CoT is generated at inference time.

## 2 RELATED WORK

**Vision-Language-Action models.** Open-source efforts in the robotics field, such as the Open X-Embodiment dataset (O'Neill et al., 2024), have fueled progress in the development of large VLAs (Kim et al., 2024; Black et al., 2024; Wen et al., 2024; Ghosh et al., 2024; Jiang et al., 2023). Recent works have also explored hierarchical VLA architectures (Shi et al., 2025; NVIDIA et al., 2025),

showing they can be beneficial for solving open-ended and long-horizon tasks. Our work aims to improve VLA's performance, by improving the way available reasoning annotations and action data is used, independently of the architecture used.

**Chain-of-Thought (CoT) and reasoning.** Generating a chain of thought has shown improved performance in LLMs solving complex reasoning tasks (Wei et al., 2022). Additionally, reasoning has shown notable success using RL with verifiable rewards, coupled with Supervised Finetuning (SFT) on example reasoning traces (DeepSeek-AI et al., 2025; Havrilla et al., 2024). CoT techniques specifically for VLMs (Shao et al., 2024) and VLAs have also been researched (Zawalski et al., 2024; Zhao et al., 2025; Lin et al., 2025; Intelligence et al., 2025). In particular, ECoT (Zawalski et al., 2024) shows that embodied thoughts can greatly improve the agent's predictions in robotics, despite the higher inference costs. Our work grounds on their findings and proposes a method that accomplishes both strong performance and fast inference.

**Hybrid reasoning.** Recent works have attempted to distill slow thinking capabilities into faster models (Deng et al., 2024; Yu et al., 2024). Closely related to our method is DualFormer (Su et al., 2025), which proposes to train a language model by systematically dropping out reasoning traces. In robotics domain, RFST (Zhu et al., 2024) proposes a hierarchical setup that uses a discriminator to decide whether to switch to the fast or slow system, with the respective model of the chosen mode being then used as the policy. Concurrent work (Chen et al., 2025) also shows that reasoning pre-training, co-training and/or dropout can improve performance of VLAs, by improving the representation learned by the model. Our work, instead, provides a method to train a single hybrid system that learns to conditionally generate a variety of outputs.

## 3   Preliminaries

Vision-Language Action (VLA) models are multimodal policies generally trained with imitation learning. A VLA processes language inputs through a Transformer-based LLM architecture (Vaswani et al., 2023; Brown et al., 2020; Team et al., 2024). Language is first "tokenized" into language tokens that are then processed by the LLM. Similarly, VLAs can process visual inputs through a vision encoder, e.g., a vision transformer (Dosovitskiy et al., 2021), that transforms image patches into visual tokens, which are then processed by the LLM.

Given a language description of a task $l$, the goal of the VLA policy is to solve the task in a given environment. The policy observes the environment through images $x$, generally captured by a camera in the environment. The policy interacts with the environment using actions $a$. Through imitation learning, the policy's objective is to learn, at each discrete timestep, the distribution $p(a_t|x_t, l)$ that solves the given task, which is empirically observed from a dataset of demonstrations.

In addition to predicting actions, thinking VLAs, like ECoT (Zawalski et al., 2024), also generate intermediate language outputs. These reasonings are expressed as thoughts $\tau$ in language form, predicted by the model. Generally, thoughts include information about the overall plan of action, the current subtask to execute, the location of objects in the image, or the direction of the agent's ongoing motion (Zawalski et al., 2024). Thinking VLA models are trained to predicts the joint probability distribution over actions and thoughts: $p(a_t, \tau_t|x_t, l) = p_\theta(a_t|x_t, l, \tau_t)p_\theta(\tau_t|x_t, l)$.

Thinking VLAs learn a single set of parameters $\theta$ to predict both actions and thoughts. Hierarchical VLAs (Shi et al., 2025; NVIDIA et al., 2025) use a two-level hierarchy of models, where one model provides an actionable language instruction for solving the task, while the second model executes the plan. By treating high-level plans and thoughts interchangeably, action prediction in hierarchical VLAs can be modelled as: $p(a_t, \tau_t|x_t, l) = p_{\theta_l}(a_t|x_t, \tau_t) \, p_{\theta_h}(\tau_t|x_t, l)$, where $\theta_h$ denotes the parameters of the "high-level" model and $\theta_l$ of the "low-level" model.

## 4   Method

VLAs that learn to predict thoughts and actions, like thinking and hierarchical VLAs, have demonstrated improved performance in several works (Shi et al., 2025; NVIDIA et al., 2025; Zawalski et al., 2024; Zhao et al., 2025). Compared to standard VLAs, these models learn to: (i) predict thoughts in language form, and (ii) condition the actions' probability on the generated thoughts. Thoughts in language form generally consist of significantly more tokens than their action counter-

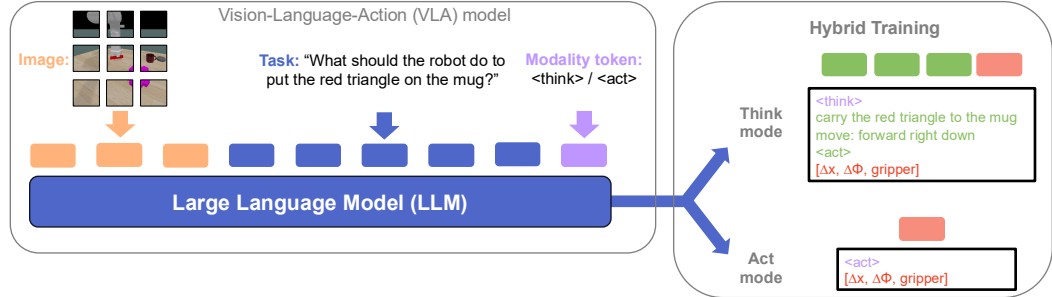

Figure 2: **Hybrid Training (HyT) framework.** Given a set of inputs, on the left, including a modality variable, the VLA model learns to conditionally generate a variety of outputs. Examples for the 'think' and 'act' conditional distributions are presented on the right.

part. Thus, generating thoughts at test-time comes at a high inference cost. This can significantly slow down the agent's action execution in the environment.

We hypothesize that the primary benefits of these models arise not from the generated thoughts themselves, but from the knowledge learned by the model through thought prediction and thought-conditioned actions prediction. This suggests that the model refines its capabilities by internalizing the patterns present in the thoughts, akin to the development of intuitive expertise (Kahneman & Klein, 2009). Under this hypothesis, after a learning process that involves thoughts conditioning and thoughts prediction, a VLA should be able to predict actions with higher accuracy, independently of the presence of thoughts as an intermediate output.

To address the need for agents capable of producing multiple probability distributions within a single model, we introduce a new training strategy, *Hybrid Training* (HyT), designed to integrate structured reasoning with flexible policy learning.

**Definition 4.1 (Hybrid Training)** *Given a task description $l$ and the current environment observation $x_t$, the conditional distribution over actions $a_t$ can be expressed as:*

$$p(a_t|x_t, l) = \sum_i \sum_j p_\theta(a_t, \tau^i, m^j|x_t, l) = \sum_i \sum_j p_\theta(a_t, \tau^i|x_t, l, m^j)p(m^j), \quad (1)$$

*by marginalizing out thoughts $\tau$ and the set of values assumed by the "modality" variable $m$.*

The hybrid training formulation allows us to describe a singla VLA model, with parameters $\theta$, that learns different conditional action distributions, mainly depending on a modality variable.

## 4.1 HYBRID TRAINING IMPLEMENTATION

The HyT formulation presented in Eq. 1 generalizes the definition of previous VLAs and enables the possibility to combine multiple objectives into a single model. In particular, leveraging the insights from other VLA models, we can use the hybrid training framework to conditionally learn three distributions:

$$p(a_t|x_t, l) = \underbrace{p_\theta(a_t|x_t, l, m^a)p_\theta(m^a)}_{\text{act}} + \underbrace{p_\theta(a_t|x_t, l, \tau_t)p_\theta(\tau_t|x_t, l, m^\tau)p_\theta(m^\tau)}_{\text{think}} + \underbrace{p_\theta(a_t|x_t, \tau_t, m^f)p_\theta(m^f)}_{\text{follow}}.$$

$$(2)$$

This way the agent will learns to generate different outputs depending on the modality variable value $m \in \{m^a, m^\tau, m^f\}$. The meaning of each distribution is defined as follows:

- **"Act" distribution**: similarly to standard VLAs, it instructs the model to directly predicts actions. It follows from $p_\theta(\tau = \emptyset|m^a) = 1$.
- **"Think" distribution**: similarly to ECoT, it instructs the model to first predict intermediate thoughts and then generate actions.
- **"Follow" distribution**: similarly to a low-level policy in a hierarchical system, it instructs the model to closely follow provided thoughts/instructions for action prediction. It follows from $p_\theta(a_t|x_t, \tau_t, m^f) = p_\theta(a_t|x_t, l, \tau_t, m^f)$ (conditional independence) and $p_\theta(\tau = \emptyset|m^f) = 1$.

Referring to our hypothesis, combining the three distributions should enable the agent to learn to output actions directly (act distribution) while also learning to predict thoughts and to follow instructions. The learning objective for the model can be described as:

$$\min_\theta \mathcal{L}_{\text{hyt}}(\theta) = w_\alpha \mathcal{L}_{\text{act}}(\theta) + w_\tau \mathcal{L}_{\text{think}}(\theta) + w_f \mathcal{L}_{\text{follow}}(\theta) \tag{3}$$

where all the terms $\mathcal{L}_\square$ are negative log-likelihood losses with respect to the corresponding outputs (actions or thoughts and actions)[1].

While one could compute the loss directly as the weighted sum of the three terms, for each datapoint, this would reduce variability in the batches, as the model would contain the same thoughts and actions multiple times in the same batch. In order to avoid this, we propose to use a Monte Carlo estimate of the objective in Equation 3. Rather than being used as loss coefficients, the weights $w_a, w_\tau, w_f$ define the probabilities of sampling the corresponding inputs and outputs for the model during training. This way, at each batch sampled during training, the model receives modality tokens, actions and thoughts, with probabilities that are defined by the coefficients.

To summarize, in order to train a VLA with HyT, one needs to specify: (i) the probability distributions to be learned, (ii) a set of values to use for the modality variable, and (iii) a set of coefficients/probabilities for sampling data during training. In this work, we adopt the distributions defined in Eq. 2, with modality variables defined as simple texts, such as "$< act >$" or "$< think >$" and with the set of coefficients $\{w_a : 0.25, w_\tau : 0.5, w_f : 0.25\}$. An ablation study on the coefficients and how each distribution impacts performance is presented in Appendix. A simplified diagram illustrating HyT (with only two kinds of outputs) is shown in Figure 2.

## 4.2    Inference Time

At test-time, the VLA is tasked to predict actions that solve a task. Thanks to HyT, it is possible to predict actions directly, leveraging the model's "act" distribution. In order to do so, we provide the model with the task description, the environments inputs, e.g. a camera image, and the modality variable $m^a =< act >$ which instructs the model to directly output actions. This enables the model to leverage the more extensive knowledge coming from learned thoughts and actions, without incurring any additional inference costs compared to standard VLAs.

While the main goal of this work is to show that HyT enables higher performance at high control frequency, the flexibility of the HyT framework also enables the possibility to instruct the agent to *operate in different "modes"*, i.e. conditionally predict different forms of outputs. Other than using the VLA in 'act' mode, it is indeed possible to use the model in 'think' or 'follow' modes.

**Q: What's the value in using other inference modalities?**

While we expect no major performance difference, e.g. when operating the model in 'think' mode rather than in 'act' mode, additional inference modes support greater flexibility in the use of the VLA. For example, the 'think' mode can be used to read the VLA's intentions, for greater interpretability. Instead, the 'follow' mode enables to provide a set of more fine-grained embodied instructions for the agent to follow, such as 'move to the left' or 'pick up the object below the gripper'. Note that, as also shown in the ECoT work (Zawalski et al., 2024), the 'thought' mode can also enable instruction-following, by overriding the agent's intentions. We verify whether these modalities can be useful with HyT in the Experiments section.

**Q: How to operate the VLA in different modalities?**

Generally, we set the modality variable to $m^a =< act >$, enforcing the model to directly predict actions. If we want to use a different mode, e.g. to predict intermediate thoughts, we can use the $m^\tau =< think >$ variable. In practice, we observed that models trained with HyT always diligently attend to the modality token and generate their outputs accordingly.

In this work, we do not explore the possibility of switching the modality token dynamically during executions, as we observed that, after HyT training, different operating modes have different generative behaviors, but similar performance. The modality variable is thus set at the beginning of a task

---

[1] For action prediction, the loss is negative log-likelihood in case the actions are discretized and predicted by the LLM. Alternatively, the loss can be an L1 or L2 loss, for continuous action prediction.

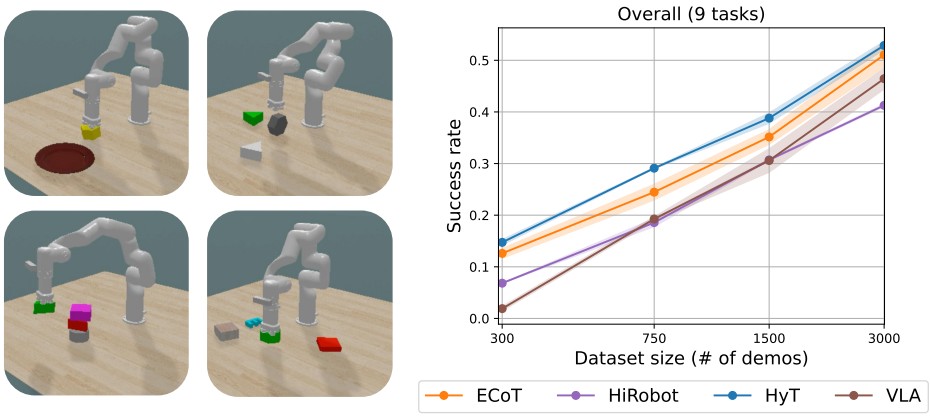

Figure 3: **ClevrSkills benchmark** Aggregated performance on 9 ClevrSkills environments (examples shown on the left). Shaded areas indicate standard errors.

execution and not updated during the episode. Further understanding and studying the tradeoffs and potential advantages that could derive from a switching mechanism between different modalities - as in (Lin et al., 2025) - is left for future work.

## 5 EXPERIMENTS

We validate the proposed HyT method in a series of simulated benchmarks, which we use to extensively assess the framework's capabilities, and on a set of real-world tasks, which demonstrates the practical utility of the approach.

**General setup.** Across all experiments, the task description is provided in the same format, i.e. *"What should the robot do to {task}?"*. Image inputs are $224 \times 224$ RGB images from a camera pointing at the robot's workspace. The action space is 7-dimensional and defined as: $[\Delta\mathbf{x}, \Delta\phi, \text{gripper}]$. The end-effector position $\mathbf{x}$ and orientation $\phi$ are controlled in delta space, while the gripper pose is in absolute value.

### 5.1 CLEVRSKILLS BENCHMARK

The ClevrSkills benchmark (Haresh et al., 2024) is based on the ManiSkill2 manipulation environments (Gu et al., 2023). The environment includes an oracle solver to collect demonstrations, and reasoning traces from the oracle, which we can use to discriminate subtasks and extract thoughts. ClevrSkills also adopts a vacuum gripper and mostly uses simple shapes and objects, isolating the challenges of manipulating complex objects, and focussing on planning and generalization instead.

**Baselines.** In this Section, we focus on comparing the HyT training methodology with other training paradigms, namely the standard VLA training (Kim et al., 2024), ECoT-like thinking VLA training (Zawalski et al., 2024), and HiRobot-like hierarchical VLA training (Shi et al., 2025).

**Dataset.** Using the oracle, we collect a diverse dataset that spans three main task groups: Place At, Place OnTop, and Stack Tower, with several variations in terms of object types and numbers. The overall dataset is made of 3000 trajectories. For all the thoughts, we adopt the same format, which includes the current subtask and the high-level motion, i.e. coarse direction instructions. This simple definition has proven effective in early stages of our analysis and in related work (Shi et al., 2025). A more detailed dataset description is given in the Appendix.

**Training.** For all approaches, we start training the VLA from the PaliGemma-2 VLM model with 3B parameters (Steiner et al., 2024), which is based on the Gemma-2 LLM (Team et al., 2024) and on the SigLIP vision encoder (Zhai et al., 2023). For action prediction, actions are tokenized using a set of 256 discrete bins, and predicted by the LLM (Kim et al., 2024). We perform full-finetuning of the model with a batch size of 32 and a learning rate of $2e-5$, using the Adam optimizer (Kingma & Ba, 2017). For the HiRobot approach, we train two distinct PaliGemma-2 models (Shi et al., 2025).

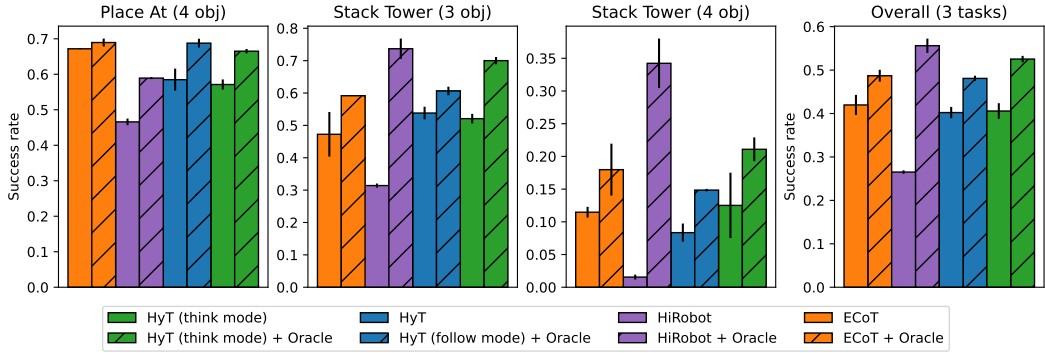

Figure 4: **Instruction-following and other inference modalities.** Comparing performance when using own generated thoughts (default) or following instructions generated by an oracle.

**Evaluation.** During evaluation, we run the agent in the environment for 100 evaluation episodes. At each episode, objects and positions are resampled randomly (but consistently among evaluation runs), making no evaluation environment exactly the same as any of the training examples. The agent is, thus, required to generalize actions to new settings in order to succeed. For each model, we evaluate 3 checkpoints taken at epochs 5, 7 and 10 during training. In general, we found no strong correlation between validation losses and performance on the task. Though, with additional training after 10 epochs, models tend to start overfitting.

**Q: How does HyT performance scales with data, compared with other training paradigms?**

With this experiment, we aim to verify the hypothesis that VLAs using HyT can develop intuitive 'thinking'. If the hypothesis is correct, we expect increased performance compared to standard VLAs, thanks to the knowledge internalized by training on reasoning traces.

As imitation learning performance can greatly vary with the number of demonstrations, we evaluate models trained with different amounts of demonstrations: 300, 750, 1500, 3000. Overall performance is summarized in Figure 3. The most evident result is that the hypothesis behind this work is confirmed: the models trained with HyT not only outperform standard VLAs, but they also generally perform better than models trained with the ECoT and HiRobot recipes, at all data scales. In Appendix, we additionally provide detailed performance per task. These results highlight that the HyT approach is particularly beneficial, compared to standard VLA training, for more complex and longer-horizon tasks.

Among the baselines ECoT performs (second) best at all data scales. Hierarchical VLAs have higher performance than standard VLAs with smaller amount of demos, but their performance increase is slower than for other methods, eventually being outperformed by standard VLAs from 1500 demos on.

In terms of inference time, we found that standard VLAs and HyT output actions at around 3Hz on A100 GPUs (4 models acting in parallel). ECoT models are $3\times$ slower. HiRobot hierarchical generation is, overall, $4\times$ slower.

**Q: How can we use the other inference modalities enabled by HyT and how do they perform?**

As described in the Method section, HyT enables the VLA to be used in multiple inference modalities, other than the 'acting' mode, directly generating actions. This can be useful in situations where interpretability or the ability to follow fine-grained instructions are useful. In order to verify empirically that these modalities are actually usable, we perform a study on the instruction following capabilities of the models on a restricted set of tasks.

In order to test instruction following from external sources, we provide the agent with a set of "Oracle thoughts", which we extract using the code from the the ClevrSkills' oracle. Oracle thoughts replace the agent's thoughts only during moving subtasks, i.e. "move to location X". This is because the oracle has precise conditions for picking and placing, compared to the VLAs. A learned policy not always satisfies them, while still being successful, causing the agent to get stuck.

Figure 5: **LIBERO benchmark** Aggregated performance on 4 LIBERO task suites (examples shown on the left). Mean performance evaluated on 100 evaluation episodes.

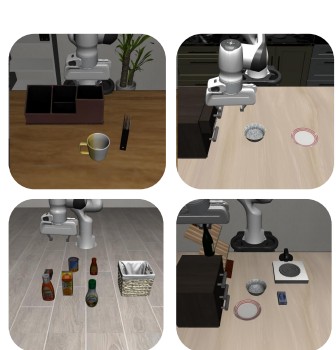

| LIBERO | Spatial | Object | Goal | Long | Avg. |
|---|---|---|---|---|---|
| *Finetuned from VLA* | | | | | |
| TraceVLA (Zheng et al., 2025) | 84.6 | 85.2 | 75.1 | 54.1 | 74.8 |
| Octo (Ghosh et al., 2024) | 78.9 | 85.7 | 84.6 | 51.1 | 75.1 |
| OpenVLA (Kim et al., 2024) | 84.7 | 88.4 | 79.2 | 53.7 | 76.5 |
| CoT-VLA (Zhao et al., 2025) | 87.5 | 91.6 | 87.6 | 69.0 | 81.1 |
| ThinkAct (Huang et al., 2025) | 88.3 | 91.4 | 87.1 | 70.9 | 84.4 |
| $\pi_0$-FAST (Pertsch et al., 2025) | **96.4** | 96.8 | 88.6 | 60.2 | 85.5 |
| MolmoAct (Lee et al., 2025) | 87.0 | 95.4 | 87.6 | 77.2 | 86.6 |
| *Finetuned from VLM* | | | | | |
| VLA-OFT (Kim et al., 2025) | 94.2 | **97.8** | 91.4 | 84.8 | 92.1 |
| **HyT** (ours) | 94.0 | 97.2 | **96.2** | **89.4** | **93.7** |

We use the models trained on the multitask dataset with the largest size (3000 demos) and present results in Figure 4 for three of the most complex tasks in the ClevrSkills benchmark adopted. HyT performance is shown in all inference modalities: 'act' mode (default, in blue), 'think' mode (in green), and 'follow' and 'think' modes with oracle thoughts (barred). First, we can observe that for all the methods tested, following instructions from an oracle, rather than self-predicted thoughts, can improve the performance of the agent. For HyT, this means that the additional inference modalities ('think' and 'follow') can actually be useful in such contexts.

Furthermore, we observe that, without oracle thoughts, models trained with HyT perform similarly in 'act' and 'think' modes, corroborating the idea that, with HyT, in the evaluated settings, intermediate thought generation at test-time may be unnecessary. Nonetheless, for future work, it would be worth verifying whether this finding holds for tasks requiring more complex embodied reasoning.

## 5.2 LIBERO BENCHMARK

The LIBERO benchmark is one of the most adopted in the VLA community (Zheng et al., 2025; Ghosh et al., 2024; Kim et al., 2024; Zhao et al., 2025; Huang et al., 2025; Pertsch et al., 2025; Lee et al., 2025; Kim et al., 2025). The agent is evaluated on 4 suites of tasks: Spatial, Object, Goal and Long (10).

In LIBERO, oracle thoughts are not available. Thus, to extract CoT, we first use the simulator to obtain labelled object bounding boxes and high-level motion primitives, including the gripper changes. Then, we feed this information step-wise, along with the task description, to an LLM that generates a plan made of subtasks and associate subtasks with temporal steps. More information about the thoughts extraction and structure can be found in the Appendix.

**Q: Can HyT be employed in combination with different VLA designs and how does performance compare to other VLAs in the literature?**

Previous work on LIBERO has shown that there are some important choices in the VLA design that are crucial to obtain high performance, such as the adoption of action chunking (Lee et al., 2025; Zhao et al., 2025; Pertsch et al., 2025; Kim et al., 2025). For this reason, we chose to implement HyT in combination with the OFT fine-tuning recipe (Kim et al., 2025), which employs action chunking and continuous actions prediction (L1 head).

For both the VLA-OFT model and HyT, we fine-tune the model starting from the Prismatic VLM (Karamcheti et al., 2024). This allows us to better leverage the language pre-training of the VLM, as also shown in (Zawalski et al., 2024). Hyperparameters and training settings are the same as in (Kim et al., 2025). Results are presented in Figure 5.

Compared to other fine-tuning recipes, the OFT strategy leverages data efficiently, outperforming all other baselines in all suites of tasks but LIBERO Spatial. Training the model with HyT allows the model to increase performance even further, especially in the most complex suites of tasks: Goal and Long. This shows that HyT can be successfully combined with well-established VLA designs, potentially improving state-of-the-art performance.

Table 1: **Real-world experiments.** Success rates with standard error on the real-world tasks (on the left). Additional details about experimental settings are provided in the Appendix.

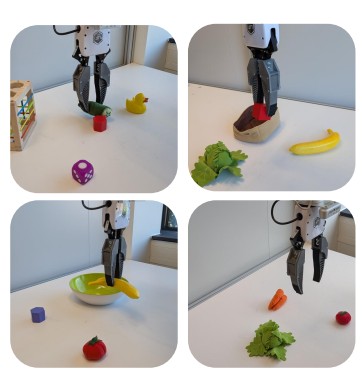

| | OpenVLA | HyT |
|---|---|---|
| **In-distribution tasks** | **52%** ±10 | **72%** ±9 |
| banana in green bowl | 70% ±15 | 70% ±15 |
| red cube in brown bag | 50% ±20 | 100% ±0 |
| tomato left of lettuce | 60% ±22 | 60% ±22 |
| zucchini in front of green cube | 0% ±0 | 50% ±25 |
| **Out-of-distribution tasks** | **29%** ±9 | **54%** ±10 |
| rubber duck in green bowl | 40% ±15 | 20% ±13 |
| mushroom in brown bag | 0% ±0 | 100% ±0 |
| purple die left of lettuce | 0% ±0 | 50% ±25 |
| zucchini in front of red hexagon | 75% ±22 | 75% ±22 |
| **Overall** | **41%** ±7 | **63%** ±7 |

We also tested fine-tuning VLA-OFT and HyT starting from the OpenVLA pre-trained VLA model. However, in this settings, we found no difference in performance (overall success: 95.3%±0.1 ), as both methods nearly saturate the benchmark's performance. One plausible conclusion is that HyT and other Chain-of-Thought (CoT) strategies, by leveraging additional thought data during training, can compensate for the lack of robotics pre-training or the scarcity of fine-tuning data, as similarly shown in the other experimental results in this work.

## 5.3 REAL-WORLD EXPERIMENTS

One of the major benefits of the HyT approach is the capability to achieve higher performance while retaining lower inference time. This is particularly useful in real-world use-cases, where faster execution is important to carry out tasks quickly and to increase the perceived quality of the agent.

For our real-world experiments, we collected a dataset comprising 320 trajectories using a robotic setup featuring an UFactory xArm 6 with a flexible two-fingered gripper, operating on a white tabletop. The agent observes the environment through RGB images captured by a RealSense D435 camera, positioned at a corner of the table. For the models, we start from the pre-trained OpenVLA model with 7B parameters (Kim et al., 2024) and compare directly to it. We perform LoRA fine-tuning with rank 32 (Hu et al., 2021), a batch size of 32 and a learning rate of $5e - 4$, using the Adam optimizer (Kingma & Ba, 2017).

Our evaluation spans two categories of tasks: in-distribution and out-of-distribution. The in-distribution set includes tasks for which the dataset contains at least 10 demonstrations. For the out-of-distribution set, we modify the in-distribution tasks with alterations, such as different objects or placements, ensuring the agent encounters novel scenarios not present in the training data. See the Appendix for additional details.

The results, shown in Table 1, show that HyT overall outperforms OpenVLA, especially in out-of-distribution tasks. From a qualitative perspective, we notice that OpenVLA and HyT have similar flaws, e.g., they tend to pick objects with the wrong orientation. However, HyT tends to be more precise when reaching picking and placing positions, e.g. it never reached for the wrong object while OpenVLA did, eventually leading to a noticeable performance gap.

## 6 DISCUSSION

Thinking strategies for VLAs (Zawalski et al., 2024; Shi et al., 2025) have shown important benefits in terms of performance and interpretability over standard VLAs, with the drawback of slower inference. In this work, we support the idea that the thinking process can be "internalized" by the model, developing some form of expert intuition (Kahneman & Klein, 2009). We proposed the Hybrid Training framework, which enables the possibility of learning from thoughts, for higher performance, while also being able to predict actions directly for faster inference, as empirically validated through simulated and real-world experiments. To conclude, we attempt to answer the question: *What is the contribution of CoT techniques to VLAs performance?*

From analyzing the HyT framework, our understanding is that learning to generate CoT and learning to predict actions from CoT improves the agent's understanding of the environment. This is closely related to how auxiliary losses have been used for imitation and reinforcement learning (Yarats et al., 2020; Srinivas et al., 2020), improving the learning dynamics of the model and generalization performance. Concurrent work (Chen et al., 2025) also supports the idea that learning from thoughts improves representation learning in VLAs.

Our work does not exclude that thoughts generation might be useful at test-time in more complex settings. Instead, it demonstrates that primitive tasks such as moving objects, opening/closing drawers, building stacks of objects, can be solved robustly and with low inference costs, by improving the model internal representations through HyT. More complex tasks requiring memory and/or advanced reasoning would still benefit from reasoning at test time. We leave the investigation of these tasks, which are currently sparse and rarely adopted in the robotics literature, for future work.

One limitation of CoT, hierarchical and HyT methods is that they require additional human labelling, providing thoughts for the agent to learn from. In this respect, one advantage of HyT approach is that it does not require the CoT to be present for the whole dataset. Instead, given that HyT relies on sampling for approximating its objective, thoughts can be sampled just for the trajectories that contain them.

Automated reasoning approaches to extract useful thoughts in a self-supervised manner would be a promising direction to explore in future work (DeepSeek-AI et al., 2025). Furthermore, we are convinced that the cost of labelling data will become negligible as state-of-the-art vision and vision-language models become more and more performant on visual and spatial reasoning tasks. In this context, HyT offers a scalable framework to distill human and foundation models' knowledge into VLA models, while strongly reducing inference costs compared to reasoning-only approaches like ECoT.

ETHICS STATEMENT

Robotic agents that are interpretable could have a positive societal impact, as they allow users to read and/or verify the agent's intention. Intervenability can both have positive and negative social impact, as it allows users to correct the agent's intentions, but it could also allow malicious users to steer the agent's behavior towards unethical behaviors.

REPRODUCIBILITY STATEMENT

In order to ensure reproducibility, we made use of open-source simulation benchmarks (ClevrSkills, LIBERO) and open-source models (Paligemma, Prismatic, OpenVLA). In the main text and/or in the Appendix, we state all the important hyperparameters for fine-tuning the models using our approach and replicate our results. We also provide details about all the tasks, chains-of-thoughts, and models' prompts employed.

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

# A   APPENDIX

## A.1   HYT COEFFICIENTS

In this Section, we investigate the impact of varying the HyT cofficients during training, using the ClevrSkills benchmark.

In Figure 6, we observe that the 'think' distribution is particularly important when the dataset available is small (300 demos). The ability to generate thoughts about the environment and the tasks may be providing a stronger training signal compared to just following detailed instructions for action prediction. As the dataset size increases, we observe that this is difference is no more present, and that models that include the 'follow' distribution actually tend to perform better.

The usefulness of the "follow" distribution is further demonstrated on the instruction-following ablation, presented in Figure 7, where we observe that despite the model trained without the 'follow' distribution has the same performance when the model is used alone, training with the 'follow' distribution improves performance when following instructions from an oracle.

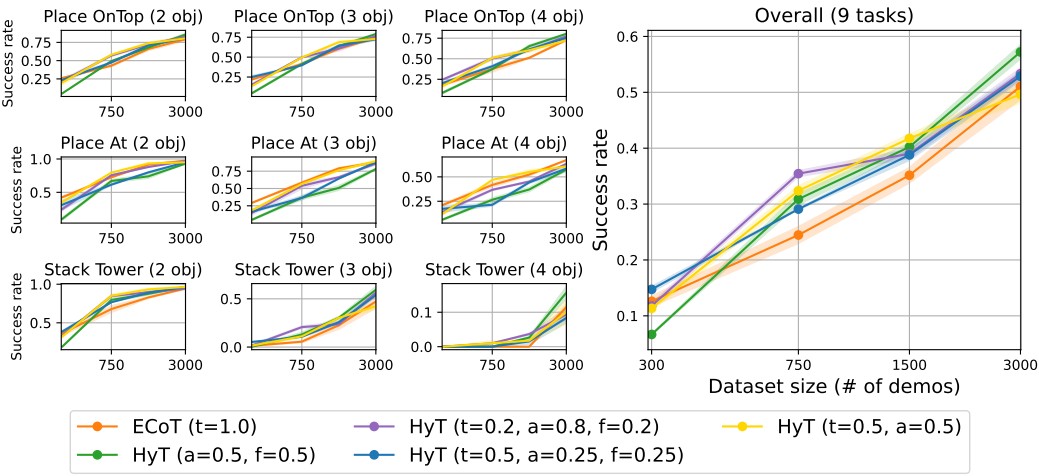

Figure 6: **Ablating HyT coefficients used during training on the ClevrSkills benchmark.**

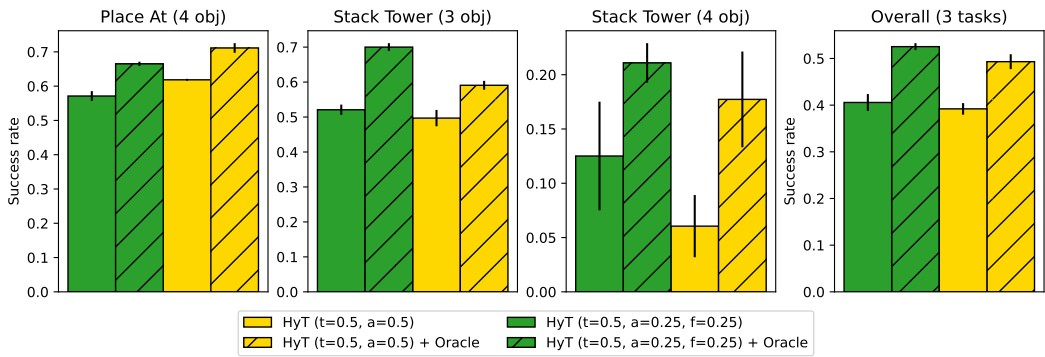

Figure 7: **Ablation on the inclusion of the 'follow' distribution during HyT training in the instruction-following benchmark.**

## A.2 CLEVRSKILLS DETAILED RESULTS

In order to provide further insights on the ClevrSkills benchmark, we provide additional results, where we grouped tasks in 3 sets:

1. *Easy tasks*: requiring only 2 actions (pick, place) and containing no distractors;
2. *Medium tasks*: requiring only 2 actions (pick, place) but containing 1+ distractors;
3. *Hard tasks*: requiring more than 2 actions or containing 2+ distractors.

For each task set, we analyze the results with 300 demos (1x data) and 3000 demos (10x data). Results are showin in Figures 8, 9, and 10.

Overall, we observe that HyT always shows large improvements over standard VLA training in the low-data regimes. When more data is provided, the performance gap is reduced for easy and medium tasks ($\sim 3 - 4\%$ difference), while it remains larger for the hard tasks ($\sim 9\%$ difference). This corroborates the idea that more complex and longer-horizon tasks benefit from training on reasoning traces more than simpler and short-horizon tasks.

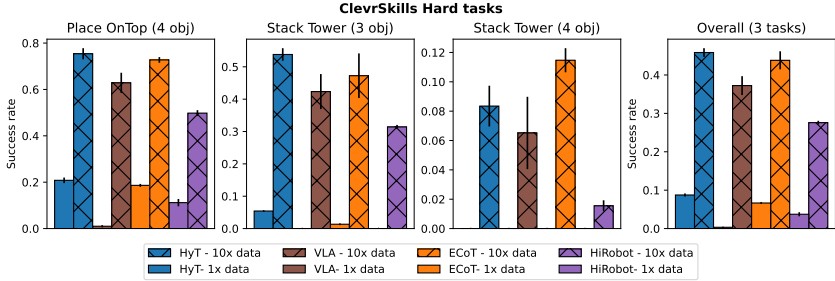

Figure 8: **Detailed results on the ClevrSkills benchmark hardest tasks.**

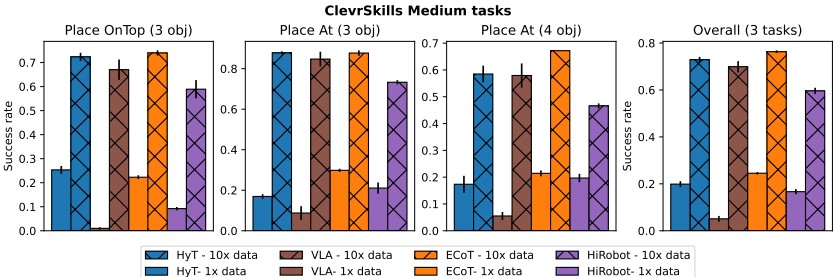

Figure 9: **Detailed results on the ClevrSkills benchmark medium tasks.**

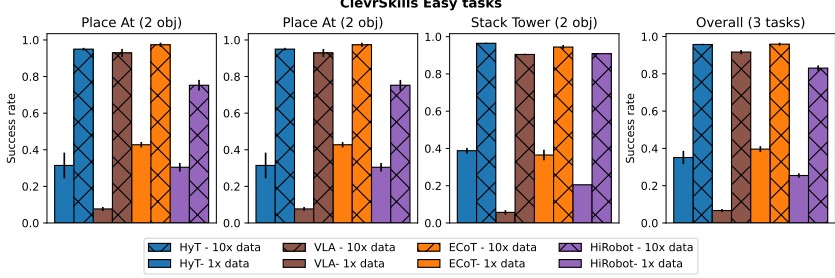

Figure 10: **Detailed results on the ClevrSkills benchmark easiest tasks.**

## A.3 CLEVRSKILLS DETAILS

**Tasks definition** The tasks are defined as follows:

- *Place At* tasks evaluate the agent's spatial understanding. The tasks require the agent to bring an object to a location specified in relation to another object location, i.e. left/right/behind/in front of the object. The objects used are mainly simple-shaped objects of different colors as in (Jiang et al., 2023).
- *Place OnTop* tasks evaluate the agent's understanding of object interactions, as they require placing an object on top of another in a stable position. The set of objects used contains simple shapes, but also more complex objects from YCB, such as mugs, wooden blocks, bowls, etc.
- *Stack Tower* evaluates the long-horizon capabilities of the agent, as it requires multiple "place on top" with simple object shapes, where the order of the objects in the stack is defined in the prompt.

For each task, we train and test the agent with three variants, containing varying number of objects. In *Place OnTop* and *Place At* the number of additional objects is only distracting or increasing the amount of clutter in the scene, e.g. making placing objects on the table harder. For the *Stack Tower* tasks more objects also require more actions, as the agent should stack all the objects present in the scene.

**Training details.** The training for all experiments and approaches is done using PyTorch DDP (Paszke et al., 2017) on 4 A100 GPUs. Inference requires less than 20GB VRAM and is performed on a multi-instance A100 for simulated environments.

**Evaluation details** At each episode, objects and positions are resampled randomly (but consistently among evaluation runs), making no evaluation environment exactly the same as any of the training examples. The agent is, thus, required to generalize actions to new settings in order to succeed. For each model, we evaluate 3 checkpoints taken at epochs 5,7 and 10 during training. In general, we found no strong correlation between validation losses and performance on the task. Though, with additional training after 10 epochs, models tend to start overfitting.

**Dataset definition.** For each task group, we collected a set of 1000 demo trajectories composed as follows: 250 demos in the 2 objects task, 500 demos in the 3 objects task, 250 demos in the 4 objects task. Then, to train agents with different sizes of the dataset, we subsample fixed subsets of trajectories from each dataset. For the multitask dataset, the task group datasets are subsampled and then aggregated.

**Extracting thoughts.** In ClevrSkills' demonstrations (Haresh et al., 2024) a variety of solvers is applied to the task, following a pre-specificied order - the oracle's plan. Each solver takes executes one subtask from the overall plan and the solver parameters can be recovered in the demonstrations. In order to create thoughts, we can use the ClevrSkills library to transform each subtask from the solver into a natural language instruction, e.g. "Move to X" or "Pick up Y". Then, for moving instructions - e.g. "Move to..." and "Carry Z to..." - we extract the motion direction of the agent. Similarly to (Shi et al., 2025), this is obtained at each timestep by computing the distance between the current end-effector position and the position at the end of the motion subtask. This is then transformed into language, in the form of "left/right", "forward/backward" and "up/down" instructions, as in (Zawalski et al., 2024). Finally, for distances that are less than 1 cm the agent receives a "close" moving instruction. Two examples of thoughts and actions, in the HyT format, are provided in Figure 11.

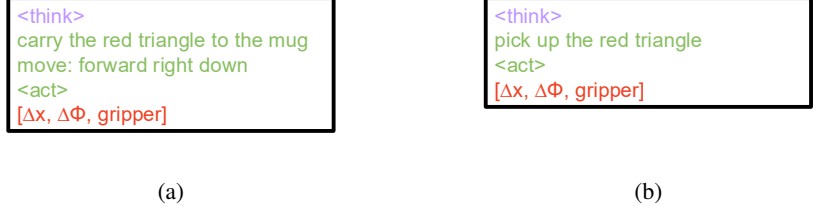

(a)               (b)

Figure 11: Examples of thoughts and actions for a moving (a) and a non-moving (b) subtasks, expressed in language format following the format used for HyT.

**Gripper information.** The ClevrSkills benchmark adopts a vacuum-gripper robot which creates a vacuum in the gripper's suction cups to maintain contact with objects. Compared to a fingers-based gripper, a suction gripper's state, i.e. open or closed, cannot be easily assessed through images. Thus, in order for the agent to be able to perform grasping actions reliably, we concatenate the state of the gripper in language form to the task description (or to the thoughts for the hierarchical low-level VLA). The gripper state information we pass to the agent is: (i) whether the gripper suction cups are making contact with something, (ii) whether the gripper was on, grasping something, or not. There's no need for the gripper information in the real-world experiments, as we use a fingers-based gripper, where the gripper state is visually observable.

## A.4 LIBERO DETAILS

**CoT extraction.** In order to generate plans and subtasks we use the Gemma-2 9B model.

For plan generation we use the following prompt:

```
I want to provide a robotic arm with a short list of high-level steps to
    complete the task: '{instruction}'.
The robotic arm is operating in a scene that contains the following
    objects:
{object_1}, {object_2}, ..., {object_n}

Break down the instruction into a Python list of high-level steps. These
    steps describe the coarse primitive actions that the robot needs to
    take to complete the task.

It is important not to invent subtasks unless they are explicitly
    required by the instruction.
For example, if the task does not mention opening or closing a drawer, do
     not include those actions.
After listing the steps, please provide a brief explanation for each one
    of the subtasks.
```

Listing 1: Prompt for LLM to generate high-level robotic steps

For inferring step-wise subtasks we use the following prompt:

```
I want to analyze each step my robot took to accomplish the task: '{
    instruction}'.
To complete this task, it followed these subtasks in order:
{subtask 1}
{subtask 2}
...
{subtask n}

Below is a mapping from each timestep to the motion the robot executed.
    The final item in each entry represents the object closest to the
    gripper. This information is used solely for reasoning purposes and
    is not part of the motion itself:
{...}

Please provide a Python dictionary that maps each of the subtasks (as
    strings) to the integer timestep at which that subtask begins. The
    mapping should include only the subtasks listed above. Since the
    subtasks are sequential, later subtasks must correspond to later
    timesteps. Each subtask must be assigned a single timestep, and the
    first subtask should always begin at step 0.
```

Listing 2: Prompt for LLM to associate subtasks with steps in an episode

The final CoTs look like this:

```
PLAN: locate the cabinet, position above the middle drawer, open the
    drawer
VISIBLE OBJECTS: akita_black_bowl_1: [138,139,140,150], cream_cheese_1:
    [157,147,170,166], wine_bottle_1: [104,96,116,122], plate_1:
    [99,171,141,196], wooden_cabinet_1: [0,94,81,212], flat_stove_1:
    [161,87,213,137], wine_rack_1: [13,27,95,123]
SUBTASK REASONING: the robot needs to move into the correct spatial
    location relative to the cabinet
SUBTASK: position above the middle drawer
MOVE: move left
GRIPPER POSITION: [106, 149]
```

Listing 3: CoT for LIBERO experiments

## A.5 REAL-WORLD EXPERIMENTS DETAILS

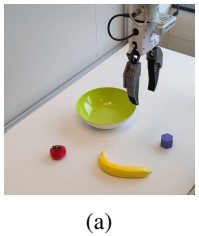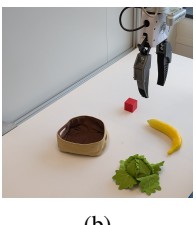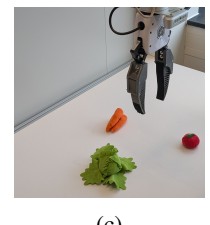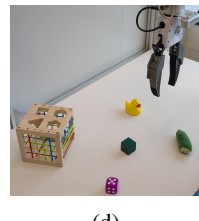

(a)       (b)       (c)       (d)

Figure 12: The setup for some of the real-world tasks: (a) banana in green bowl (b) red cube in brown bag (c) zucchini in front of green cube (d) tomato left of lettuce.

**Dataset preparation.** The dataset for real-world experiments is collected via human tele-operation, using a VR set and a controller to map the human motions and intentions, e.g., closing the gripper, to the robot. After collecting the trajectories, we apply some basic pre-processing operations. First, we take the sequences collected at 60 Hz, and we subsample them at 10 Hz. Then, we extract actions in the format $[\Delta x, \Delta \phi, \text{gripper}]$ from the end-effector and gripper positions of the demonstrations. Finally, we filter out no-motion operations, i.e. having $\Delta x$ and $\Delta \theta$ equal to zero.

**Extracting thoughts.** For extracting the subtasks to use in the thoughts, we use a simple heuristic-based approach. This is based on the assumption that all the tasks in the dataset are in the format 'place the {obj1} {position} the {obj2}', e.g. 'place the {zucchini} {in front of} the {green cube}'. We identify the position when the robot closes the gripper as the *grasping position*, and the position when the robot opens the gripper as the *releasing position*. Then, we identify the following 3 keyframe moments: (i) the moment when the agent reaches within 3 cms distance from the grasping position, (ii) the grasping moment, (iii) the moment when the agent reaches within 3 cms distance from the releasing position. Finally, the subtasks are defined in the trajectory as follows:

- *Move to the {obj1}*: between the start of the trajectory and keyframe (i);
- *Pick up the {obj1}*: between keyframe (i) and keyframe (ii);
- *Move {position} the {obj2}*: between keyframe (ii) and keyframe (iii);
- *Place object {position} the {obj2}*: between keyframe (iii) and the end of the trajectory.

Also, to have the same thought structure as in the ClevrSkills' experiments, we concatenate a 'move: {direction}' to the moving subtasks, where the direction is computed using the end-effector position (see Section A.3).

**Evaluation.** During evaluation, we limit the execution time to 150 agent steps, which is $\sim 2$ minutes per trajectory, considering 3 actions/second from the model and a control loop running at $\sim 2$Hz for stable motions. In case of failed grasps, we allow up to 3 attempts to the agent, before assessing the outcome of the episode. In the following, you find a detailed description of how each in-distribution task is executed and randomized. For out-of-distribution tasks, we follow the same procedures, but we swap one of the main actors in the scene (e.g. the object to pick or the placing target). See Figure 12) for reference. Note: fruits are stuffed toys, while vegetables are hard foam models.

- *Place the banana in the green bowl (10 trials).* The scene includes a green bowl and three objects: a banana, a tomato, and a blue hexagon, each with predefined positions. We conduct five trials with the banana starting in one location, and five more from a different location. In each location, the banana appears in three orientations: vertical (2/5 trials), horizontal (2/5 trials), and diagonal (1/5 trial).
- *Place the red cube in the brown bag. (6 trials).* The setup features a brown bag and three objects: a red cube, a banana, and a lettuce leaf, all placed at specific locations. We run two trials for each location of the red cube. Its orientation varies between being aligned with the robot's base and being tilted.
- *Place the tomato left of the lettuce (5 trials).* The scene contains three objects: a lettuce leaf, a carrot, and a tomato. Their positions are randomized within defined regions, though the overall layout remains consistent. We perform five trials with these randomized placements. A trial is only considered successful if the tomato ends up on the table, near the lettuce, and to its left.
- *Place the zucchini in front of the green cube (4 trials).* The environment includes five objects: a shape sorting box, a zucchini, a purple die, a rubber duck, and a green cube. Object positions are randomized within certain bounds, while maintaining the general layout. We carry out four trials with these randomized setups. Success is defined as the zucchini being placed on the table, close to and in front of the green cube.

