# OpenReview forum: "Hybrid Training for Vision-Language-Action Models"
_ICLR.cc/2026/Conference — ICLR 2026 Poster_

### Official Review · Reviewer_LZtX · 2025-10-31

**Soundness:** 3
**Presentation:** 3
**Contribution:** 3
**Rating:** 6
**Confidence:** 4

**Summary:**

This paper proposes Hybrid Training (HyT), a framework that lets VLAs learn from CoT reasoning for better performance while allowing inference without generating thoughts. HyT flexibly enables direct action prediction, thought generation, or instruction following, and shows strong results in both simulation and real-world experiments.

**Strengths:**

1. Performance-speed breakthrough: HyT resolves the key “reasoning vs. reflex” trade-off in VLAs, matching the success rates of CoT methods while maintaining high inference speed (e.g., 3 Hz vs. 1 Hz for ECoT)—a major step toward practical deployment.

2. Robust empirical results: Extensive experiments show HyT’s clear advantages: strong data-scaling on ClevrSkills, state-of-the-art results on LIBERO (especially “Goal” and “Long” tasks), and real-world robot transfer. Its 54% OOD success vs. 29% baseline highlights superior generalization.

3. Versatile and interpretable model: A single HyT-trained model supports fast “act,” interpretable “think,” and guided “follow” modes—allowing deployment, analysis, and human-robot collaboration without retraining.

**Weaknesses:**

1.  HyT requires CoT or reasoning traces for training—using oracle thoughts on ClevrSkills and Gemma-2 9B-generated plans on LIBERO. This reliance on costly, curated reasoning data limits scalability and makes adaptation to new domains harder than learning directly from demonstrations.

2. While “act” mode matches “think” mode performance, the paper doesn’t test harder, long-horizon, or abstract tasks where explicit reasoning might still be needed. Without such analysis, the boundaries of HyT’s internalized reasoning remain uncertain.

**Questions:**

All of my qeustions are listed in the weakness section. If my concerns are well addressed, I will raise my rating.

---

> ### Author Response · Authors · 2025-11-21
>
> We thank the reviewer for their useful feedback. We have updated the manuscript to answer some of the reviewers' concerns, using the blue text for additions and the green text for edits. We provide responses to this reviewer's concerns as follows.
>
> > HyT requires CoT or reasoning traces for training—using oracle thoughts on ClevrSkills and Gemma-2 9B-generated plans on LIBERO. This reliance on costly, curated reasoning data limits scalability and makes adaptation to new domains harder than learning directly from demonstrations.
>
> As stated in the Discussion, we agree that annotating data with embodied reasoning can be costly. However, we are convinced that this cost will become negligible as state-of-the-art vision and vision-language models become more and more performant on visual and spatial reasoning tasks. The performance of models like SAM 3 [1] is already approaching human's proficiency at identifying and localizing objects in visual scenes. Models like Gemini 3 Pro [2] offers advanced visual reasoning capabilities, as showcased in benchmarks like MMMU-Pro where frontiers models performance has been steadily improving over time (https://llm-stats.com/benchmarks/mmmu-pro).
>
> In this context, HyT offers a scalable framework to distill human and foundation models' knowledge into VLA models, while strongly reducing inference costs compared to reasoning-only approaches like ECoT.
>
> > While “act” mode matches “think” mode performance, the paper doesn’t test harder, long-horizon, or abstract tasks where explicit reasoning might still be needed. Without such analysis, the boundaries of HyT’s internalized reasoning remain uncertain.
>
> As stated in the Introduction, our main hypothesis is that using HyT the model can internalize most of the information contained in the reasoning traces, similarly to the way humans develop "skilled intuition". In this work, we showed that this applies successfully to primitive tasks such as moving objects, opening/closing drawers, building stacks of objects, which can be solved robustly and with low inference cost without reasoning. However, as stated in the Discussion, "we do not exclude that thoughts generation might be useful at test-time in more complex settings". In particular, we agree that tasks that require more advanced reasoning such as "solve a math equation and write down the result on a piece of paper" or "clean up the kitchen, putting all items back to their original place" will benefit from interleaving reasoning and actions at test-time.
>
> In the updated manuscript, we edited our statements in the Discussion to more accurately reflect our perspective on the matters here commented and improve clarity.
>
> [1] SAM 3: Segment Anything with Concepts, Carion et al, 2025
>
> [2] Gemini 3 Pro, Google, 2025, https://deepmind.google/models/gemini/pro/
>
> ---
>
> We would like to thank the reviewer once again for contributing to the review process of our paper. We hope to have address their  concerns thoroughly and we look forward to receiving an updated feedback.

---

### Official Review · Reviewer_zK8v · 2025-11-01

**Soundness:** 3
**Presentation:** 3
**Contribution:** 2
**Rating:** 4
**Confidence:** 3

**Summary:**

This paper proposes Hybrid Training (HyT), a method that unifies thinking and acting behaviors in a single vision-language-action model via a learnable modality variable controlling “think/act/follow” modes. HyT trains one model to generate both reasoning traces and executable actions, using a Monte-Carlo sampling scheme to mix modalities during training. The method is evaluated across simulated (ClevrSkills, LIBERO) and real-robot (xArm6) environments, demonstrating competitive task success and faster inference compared to full chain-of-thought reasoning.

**Strengths:**

1. HyT uses a modality variable to avoid maintaining separate reasoning and acting modules, simplifying deployment and enabling run-time mode control (think/act/follow) without model-switching.
2. HyT shows good performance on the ClevrSkills benchmark, suggesting HyT can learn complex, compositional behaviors required there. And robotic experiments on an xArm6 support practical applicability beyond simulation.

**Weaknesses:**

1. In Figure 4, reported task success and efficiency gains relative to ECoT are small; it remains unclear whether HyT achieves comparable performance at lower latency or merely approximates ECoT under similar compute.
2. How does HyT compare to the π₀.₅ model (https://github.com/Physical-Intelligence/openpi) and OneTwoVLA (https://arxiv.org/pdf/2505.11917)? Can the BOA/BOR control mechanism (as in OneTwoVLA) achieve similar adaptive switching without retraining?

**Questions:**

See Weaknesses.

---

> ### Author Response · Authors · 2025-11-21
>
> We thank the reviewer for their useful feedback. We have updated the manuscript to answer some of the reviewers' concerns, using the blue text for additions and the green text for edits. We provide responses to this reviewer's concerns as follows.
>
> > In Figure 4, reported task success and efficiency gains relative to ECoT are small; it remains unclear whether HyT achieves comparable performance at lower latency or merely approximates ECoT under similar compute.
>
> We agree that HyT only slightly outperforms ECoT in terms of success rate. However, in contrast with the reviewer's concern, the efficiency gains are very significant. As shown in the ClevrSkills benchmark (see Figure 1), HyT has a 3x faster control cycle. As also recognized by all the other reviewers, this is a large compute effiency gain that is greatly desirable in robotic manipulation applications (Strength 1 for Rev. cYRE, Strength 2 for Rev. PRFm, Strength 1 for Rev. LZtX).
>
> > How does HyT compare to the π₀.₅ model (https://github.com/Physical-Intelligence/openpi) and OneTwoVLA (https://arxiv.org/pdf/2505.11917)?
>
> Both $\pi_{0.5}$ and OneTwoVLA incorporate reasoning data during training, training the model to be able to output both semantic information about the environment (System II-like), and actions conditioned on this information (System I-like). In particular, $\pi_{0.5}$ during each step of inference first predicts a short reasoning trace, and then predicts the low-level robot action chunk. Instead, the OneTwoVLA model decides at each step whether to reason (BOR token) or act (BOA token). This is achieved by defining two intervals in the data trajectories: reasoning intervals, capturing key steps requiring model reasoning, and acting intervals, in which the model primarily learns to predict actions based on observations and the latest reasoning content.
>
> Our work is based on a different hypothesis, which leads to a very different method from the $\pi_{0.5}$ and OneTwoVLA works. As stated in the Introduction, our main hypothesis is that using HyT, learning different conditional action probabilities, the model can internalize most of the information contained in the reasoning traces, similarly to the way humans develop "skilled intuition". By doing so, the model improves its internal representation of the environment and can obtain higher performance without generating any reasoning at test-time. We showed that this applies successfully to primitive tasks such as moving objects, opening/closing drawers, building stacks of objects, which can be solved robustly and with low inference costs without reasoning.
>
> > Can the BOA/BOR control mechanism (as in OneTwoVLA) achieve similar adaptive switching without retraining?
>
> As stated in Section 4.2, it is possible to switch modalities during inference, without retraining. However, in this work, we do not explore this possibility. This is because we observed that, after HyT training, different operating modes have different generative behaviors but similar performance. The modality variable is thus set at the beginning of a task execution and not updated during the episode. Further understanding and studying the tradeoffs and potential advantages that could derive from a switching mechanism between different modalities (e.g. as done in OneTwoVLA) is left for future work.
>
> We acknowledge the importance of recognizing concurrent efforts in the field and we have updated our Related Work to include references to the mentioned works. However, we would like to point out that both $\pi_{0.5}$ and OneTwoVLA are very recent and concurrent works. $\pi_{0.5}$ was published at CORL 2025 - accepted on August 1st and presented in late September (after the ICLR deadline). OneTwoVLA is under review for this same conference - ICLR 2026 submission: https://openreview.net/forum?id=tWMfhoP3as&noteId=9mDjjHyPRy.
>
> ---
>
> We would like to thank the reviewer once again for contributing to the review process of our paper. We hope our comments provide clarity on the weaknesses pointed out by the reviewer and we look forward to receiving an updated feedback.

---

### Official Review · Reviewer_PRFm · 2025-11-01

**Soundness:** 3
**Presentation:** 4
**Contribution:** 3
**Rating:** 6
**Confidence:** 4

**Summary:**

This paper introduces Hybrid Training (HyT), a framework for training Vision-Language-Action models that enables learning from chain-of-thought (CoT) reasoning traces while maintaining fast inference speeds. The key insight is that VLAs can internalize knowledge from thought supervision during training without needing to generate thoughts at test time. HyT trains a single model to conditionally predict three distributions based on a modality variable: direct actions ("act"), thoughts then actions ("think"), and following provided instructions ("follow"). The method uses Monte Carlo sampling during training for the three modes. Experiments on ClevrSkills, LIBERO, and real-world tasks demonstrate that HyT achieves performance gains similar to ECoT while maintaining standard VLA inference speeds (3Hz vs 1Hz for ECoT).

**Strengths:**

- The hypothesis that models can internalize CoT reasoning without explicit generation is compelling and well-articulated through the System 1/System 2 cognition analogy.
- HyT outperforms competitive baselines including MolmoAct (86.6% → 93.7%) and π0-FAST (85.5% → 93.7%) on LIBERO, while maintaining 3× faster inference than ECoT.
- The paper includes extensive experiments across simulated (ClevrSkills, LIBERO) and real-world environments with ablations on data scaling.
- The ability to maintain ECoT-level performance at standard VLA inference speeds addresses a critical deployment constraint for robotic systems.

**Weaknesses:**

- The loss coefficients (wa:0.25, wτ:0.5, wf:0.25) appear arbitrary without ablation
- The Monte Carlo sampling approach is not compared to direct weighted loss computation
- Individual contributions of Lfollow and Lthink to overall performance are not analyzed
- HiRobot with oracle thoughts often outperforms HyT with oracle, suggesting the "follow" mode may not be optimally implemented.
- The paper lacks systematic investigation of which task characteristics benefit most from internalized CoT training versus standard VLA training.
- Unclear if all baselines use the same VLM backbone (PaliGemma-2)
- No inference speed comparison with π0-FAST (or other baselines)
- Compatibility with other VLA architectures not explored

**Questions:**

- Why does HiRobot+Oracle outperform HyT+Oracle in Figure 4?
- How were the loss coefficients (0.25, 0.5, 0.25) determined? What is the sensitivity to these values?
- What is the ablation comparing Monte Carlo sampling versus direct weighted loss computation? Does the sampling variance help or is it just for computational efficiency?
- Can you provide ablations showing individual contributions of Lthink and Lfollow? What happens with just Lact + Lthink?
- What are the specific inference speeds compared to π0-FAST, and can HyT be combined with their efficient tokenization approach?
- In what types of tasks (complexity, horizon length, reasoning requirements) does HyT show the largest improvements over standard VLA training?
- Do all compared baselines (Table in LIBERO results) use the same PaliGemma-2 backbone for fair comparison?

---

> ### Author Response · Authors · 2025-11-21
>
> We thank the reviewer for their useful feedback. We have updated the manuscript to answer some of the reviewers' concerns, using the blue text for additions and the green text for edits. We provide responses to this reviewer's concerns as follows.
>
> > Weaknesses and questions about the loss coefficients and contributions of the different losses (W1, W3, Q2)
>
> To respond to the reviewer's questions, we have run two additional experiments, which can be found in the Appendix of the updated manuscript. Please find as follows an answer to the questions given.
>
> Q: Individual contributions of Lfollow and Lthink to overall performance are not analyzed.
>
> A: In the ClevrSkills benchmark ablation study (Figure 6), we observe that the "think" distribution is particularly important when the dataset available is small (300 demos). The ability to generate thoughts about the environment and the tasks may be providing a stronger training signal compared to just following detailed instructions for action prediction. As the dataset size increases, we observe that this is difference is no more present, and that models that include the "follow" distribution actually tend to perform better.
>
> The usefulness of the "follow" distribution is further demonstrated on the instruction-following ablation (Figure 7), where we observe that despite the model trained without the "follow" distribution has the same performance when used alone, training with the "follow" distribution improves performance when following instructions from an oracle.
>
> Q: How were the loss coefficients (0.25, 0.5, 0.25) determined? What is the sensitivity to these values?
>
> During early developments of the approach, we observed that balanced combinations of the three coefficients consistently provided strong and robust performance. This initial finding, which led to our choice of the coefficients, is further reinforced by our ablation study. In general, any application of HyT (with the exception of the model excluding the "think" distribution with 300 demos) performs in par or better than ECoT at any data scale.
>
> > Difference between Monte Carlo sampling versus direct weighted loss computation (W2, Q3)
>
> The choice of sampling according to the coefficient probabilities, rather than implementing the coefficients in the loss at every batch is convenient from a practical point of view. It allows each dataloader worker to sample different datapoints independently and it also enables one to use a dataset that does not contain reasoning traces for all examples. We do not expect major performance differences between the sampling-based and the weighted-loss implementations, though the latter is more cumbersome to adopt.
>
> > Why does HiRobot+Oracle outperform HyT+Oracle in Figure 4? (W4, Q1)
>
> Compared to HyT, HiRobot trains two different models: a high-level model, translating the task into detailed instructions, and a low-level model, translating instructions into actions. When the model is used in standard settings, this setup requires both the high-level and the low-level models to be accurate in their predictions. This design can cause the model to underperform, when both models are used in standard settings (see Figure 3).
>
> However, when oracle instructions are provided, as in the instruction-following benchmark (Figure 4), the low-level model is more specialized than the HyT one in following instructions accurately. In particular, HyT might still end up following its internalized intentions, ignoring (part of) the oracle instructions. Learning the "follow" probablity in HyT alleviates this issue, as shown in the newly presented ablations (see updated manuscript's Appendix), but we found the specialized model to still perform best in these settings.

---

> ### Author Response · Authors · 2025-11-21
>
> > Comparison and compatibility with different backbones, tokenization approaches, VLA architectures (W6,W7,W8, Q5, Q7)
>
> For all the ClevrSkills experiments, we have trained our approach and all the baselines, based on the same PaliGemma-2 VLM backbone, and using a discrete tokenizer similar to the OpenVLA tokenizer [1] - as stated in Section 5.1.
>
> For the LIBERO experiments, we have trained our approach and the VLA-OFT baseline, using the same Prismatic VLM backbone, and training a continuous action prediction head - as stated in Section 5.2. For all the other approaches, we reported the values from the MolmoAct paper [1]. For details about the VLM backbone and action prediction scheme adopted, we have provided the corresponding reference for each approach in the results' table.
>
> Responding to the reviewer's concerns, we have already showed that our approach can be successfully implemented on top of 3 different VLA architectures: (i) PaliGemma VLM + a discrete action tokenizer for ClevrSkills experiments, (ii) Prismatic VLM a continuous action prediction head in LIBERO, and (iii) OpenVLA for real-world experiments.
>
> HyT could naturally be applied in combination with the FAST tokenization approach too. While we do not have a 1:1 comparison with the same backbone and tokenizer, we can confidently state that implementing HyT with the same architecture and tokenization scheme would offer no improvements in terms of latency compared to $\pi_0$-FAST. However, HyT thanks to learning the "think" and "follow" distributions could definitely push the performance of the model further, similarly to what we saw with the VLA-OFT model in LIBERO.
>
> > In what types of tasks (complexity, horizon length, reasoning requirements) does HyT show the largest improvements over standard VLA training?
>
> As stated in Section 5.2, in the LIBERO benchmark, training the model with HyT allows the model to increase performance, especially in the most complex suites of tasks: Goal and Long.
>
> In order to provide further insights on the ClevrSkills benchmark, we have provided detailed plots in the Appendix of the updated manuscript. We grouped tasks in 3 sets:
> * *Easy tasks*: requiring only 2 actions (pick, place) and containing no distractors
> * *Medium tasks*: requiring only 2 actions (pick, place) but containing 1+ distractors
> * *Hard tasks*: requiring more than 2 actions or containing 2+ distractors
>
> For each task set, we provide details with 300 demos (1x data) and 3000 demos (10x data).
>
> Overall, we observe that HyT always shows large improvements over standard VLA training in the low-data regimes. When more data is provided, the performance gap is reduced for easy and medium tasks (around 3-4% difference), while it remains larger for the hard tasks (around 9% difference). This corroborates the idea that more complex and longer-horizon tasks benefit from training on reasoning traces more than simpler and short-horizon tasks. We also included a statement in the main section of the manuscript, in Section
>
> ---
>
> We would like to thank the reviewer once again for contributing to the review process of our paper and we remain available for further clarifications.
>
> [1] OpenVLA: An Open-Source Vision-Language-Action Model, Kim et al, 2024
>
> [2] MolmoAct: Action Reasoning Models that can Reason in Space, Lee et al, 2025

---

> > ### Comment · Reviewer_PRFm · 2025-11-25
> > **Concerns addressed**
> >
> > Thank you for the detailed response. I appreciate the clarifications. The ablations of Lfollow and Lthink are helpful to understand when each is useful. Even though there is some variation depending on the loss coefficients used, it’s nice to see they are generally robust compared to ECoT. I raised my score.

---

### Official Review · Reviewer_cYRE · 2025-11-02

**Soundness:** 2
**Presentation:** 3
**Contribution:** 2
**Rating:** 6
**Confidence:** 4

**Summary:**

This paper proposes that the benefits of chain-of-thought (CoT) reasoning in VLAs can be reaped without the very slow inference times that generating reasoning chains at test-time requires. Namely, they propose a method called HyT (hybrid training) in which they train the VLA to exhibit multiple modes of behaviors, either predicting actions directly, predicting a reasoning chain and then actions, or predicting actions from an oracle human-provided reasoning chain, with these modes selectable by configuring a modality variable. This hybrid training scheme allows the model to effectively learn from the chain-of-thought data at train time, not having to actually generate these reasoning chains at test-time to still get equally good performance. The authors test their approach against a standard VLA, a reasoning VLA, and a hierarchal VLA on the ClevrSkills simulated benchmark and find that their approach performs the best across multiple dataset sizes. On the Libero simulated benchmark, there approach, adapted off of the OFT-VLA model, performs the best when compared to many prior VLAs on Libero. And finally on eight tasks in the real-world, their model exhibits better performance than a standard VLA while requiring the same inference time (~3 Hz).

**Strengths:**

(1) The authors tackle a very relevant problem in robotics, and arrive at a nice result in that the benefits of reasoning trace generation can be obtained without having to generate expensive reasoning chains at test time. The proposed hybrid training strategy is general and permits different manners of either conditioning on or not conditioning on reasoning at test time.

(2) The experimental results are reasonably through, and include real-world results, which is nice. In particular, the authors’ method scores the highest on the Libero simulated benchmark when compared to other fine-tuned VLAs.

(3) The paper is clear and easy to read.

**Weaknesses:**

(1) The analysis of the results could be improved. In particular, it remains unclear what the specific reason is that reasoning trace generation can be avoided at test-time, but yet the benefits of reasoning can still be obtained. In their discussion section the authors wrote that “from analyzing the HyT framework, our understanding is that learning to generate CoT and learning to predict actions from CoT improves the agent’s understanding of the environment” —> could you expand on what it mean to improve the understanding of the environment? I.e., is it that the model can obtain better internal representations if trained with reasoning, or can follow language better, or something else?

(2) Reasoning is perhaps the most appealing when the task involves some complex decision making/logical inference before picking what atomic action to take. It seems that the authors missed trying their approach out on these types of tasks. Indeed from the discussion: “we do not exclude that thoughts generation might be useful at test-time in more complex settings, but this would require evaluating on tasks that require advanced reasoning capabilities, which are currently sparse and rarely adopted in the robotics literature”. It would have been interesting to see whether the hybrid training scheme can bring benefits to these types of tasks as well, or if reasoning at test-time becomes strictly necessary.

**Questions:**

(1) For the three different modes, the authors chose weights 0.25 for act, 0.5 for think, and 0.25 for follow. Did they ablate different values of these weights? Generally how important is a good selection of these weights to model performance? Which of the three modes matter the most/least?

(2) HiRobot + Oracle seems to be doing the best, as shown in Figure 4. Am I correct in assuming that the main difference between HiRobot and the proposed approach is that two separate PaliGemma models are used? What leads to the improved performance? More capacity? Less feature sharing?

(3) Why do you think ECoT performed worse than the proposed approach? They are very similar, and it would seem that ECoT should perform slightly better since it generates reasoning at test-time?

(4) Why aren’t any of the ClevrSkills baselines compared against in Libero? E.g., ECoT or HiRobot?

(5) For the real-world tasks, what do the reasoning traces look like and how were they generated? Were the types of reasoning traces generated such that reasoning might be particularly useful in the OOD setting?

(6) There appears to be substantial overlap with a contemporaneous, peer-reviewed publication ([1], accepted Aug 1, 2025). Per the ICLR 2026 reviewer FAQ, I will not penalize the paper for missing a citation or comparison to contemporaneous work. However, as written, the submission does not clearly articulate contributions beyond that work. It would be great if the authors could clarify what is genuinely new here (theory, method, or empirical evidence) beyond this prior work.

[1] Chen, W., Belkhale, S., Mirchandani, S., Mees, O., Driess, D., Pertsch, K., & Levine, S. (2025). Training Strategies for Efficient Embodied Reasoning. arXiv preprint arXiv:2505.08243.

---

> ### Author Response · Authors · 2025-11-21
>
> We thank the reviewer for their useful feedback. We have updated the manuscript to answer some of the reviewers' concerns, using the blue text for additions and the green text for edits. We provide responses to this reviewer's concerns as follows.
>
> > The analysis of the results could be improved. (...). Could you expand on what it mean to improve the understanding of the environment? (W1)
>
> We agree with the reviewer's comment that learning to model multiple conditional action probabilities with HyT improves the model's internal representations, which is what we intend to convey when saying that HyT improves the model's understanding of the environment. For example, learning to predict the current subtask forces the model to learn to identify the patterns that visually separate subtasks, such as carrying/picking/dropping an object. We believe this is the key reason of success behind works like ECoT, rather than the presence of the thoughts in the agent's context during inference. We based our work on this hypothesis, and show that this holds across several settings.
>
> > Reasoning is perhaps the most appealing when the task involves some complex decision making/logical (...). It would have been interesting to see whether the hybrid training scheme can bring benefits to these types of tasks as well (...) (W2)
>
> In this paper, we focussed on the idea that test-time reasoning is not really required for most of the tasks VLAs are generally used for, which involve simple combinations of primitive tasks, such as moving objects, opening/closing drawers, building stacks of objects, which can be solved robustly and with low inference cost without reasoning. We agree that tasks that require more advanced reasoning such as "solve a math equation and write down the result on a piece of paper" or "clean up the kitchen, putting all items back to their original place" will benefit from interleaving reasoning and actions at test-time. Nonetheless, we agree with the reviewer that it would be interesting to see whether HyT could still beneficial in these settings or at least to solve simpler reasoning tasks.
>
> We have updated the related paragraph in the Discussion of the manuscript to better reflect our perspective on this matter.
>
> > Ablations of coefficients during training (Q1)
>
> To respond to the reviewer's questions, we have run two additional experiments, which can be found in the Appendix of the updated manuscript. Please find as follows an answer to the questions given.
>
> Q: Which of the three modes matter the most/least?
>
> In the ClevrSkills benchmark ablation study (Figure 6), we observe that the "think" distribution is particularly important when the dataset available is small (300 demos). The ability to generate thoughts about the environment and the tasks may be providing a stronger training signal compared to just following detailed instructions for action prediction. As the dataset size increases, we observe that this is difference is no more present, and that models that include the "follow" distribution actually tend to perform better.
>
> The usefulness of the "follow" distribution is further demonstrated on the instruction-following ablation (Figure 7), where we observe that despite the model trained without the "follow" distribution has the same performance when used alone, training with the "follow" distribution improves performance when following instructions from an oracle.
>
> Q: How important is a good selection of these weights to model performance?
>
> A: From our ablation study, it emerges that balanced combinations of the three coefficients provide both high performance and good instruction-following capabilities. In general, any application of HyT (with the exception of the model excluding the "think" distribution with 300 demos) performs in par or better than ECoT at any data scale.
>
> > HiRobot + Oracle seems to be doing the best, as shown in Figure 4. (...) (Q2)
>
> As correctly pointed out by the reviewer, compared to HyT and ECoT, HiRobot trains two different models: a high-level model, translating the task into detailed instructions, and a low-level model, translating instructions into actions. When the model is used in standard settings, this setup requires both the high-level and the low-level models to be accurate in their predictions. This design can cause the model to underperform, when both models are used in standard settings (see Figure 3).
>
> When oracle instructions are provided, as in the instruction-following benchmark (Figure 4), the low-level model is more specialized than the HyT and ECoT in following instructions accurately. In particular, HyT and ECoT might still end up following their internalized intentions, ignoring (part of) the oracle instructions. Learning the "follow" probablity in HyT alleviates this issue, as shown in the newly presented ablations (see updated manuscript's Appendix), but the specialized model still performs best in these settings.

---

> > ### Author Response · Authors · 2025-11-21
> >
> > > Why do you think ECoT performed worse than the proposed approach? (Q3)
> >
> > ECoT is trained to predict reasoning traces conditioned on the task, and actions conditioned on the tasks and the reasoning traces. While this improves the representation learned by the model, it also makes so that the action's accuracy is also conditional on the thought's accuracy. Failing to accurately predict reasoning traces, may lead the model to predict the wrong actions.
> >
> > In HyT, this problem is alleviated as: (i) the model learns multiple conditional probabilities, which further enchances the model's internal representation, (ii) the model requires no reasoning trace prediction during inference time. HyT also reduces the risk to learn spurious correlations from the provided reasoning traces, an issue that is related to causal confusion in imitation learning [1].
> >
> > > Why aren’t any of the ClevrSkills baselines compared against in Libero? E.g., ECoT or HiRobot? (Q4)
> >
> > We presented a systematic evaluation of different training strategies in the ClevrSkills benchmark, which allowed us to flexibly study performance at different data scales.
> >
> > For the LIBERO benchmark, we focussed on answering two main questions: (i) can HyT be combined with other VLA architectures and consistenly improve performance?, (ii) can we use HyT to improve performance that are close to the state-of-the-art?
> >
> > For answering these questions, in order to reduce the overall computational costs of this work, we only trained our baseline implementation (VLA-OFT) and our approach, to provide a fair comparison. For the rest, we used the baselines scores reported in the MolmoAct paper [2]
> >
> > > For the real-world tasks, what do the reasoning traces look like and how were they generated? Were the types of reasoning traces generated such that reasoning might be particularly useful in the OOD setting? (Q5)
> >
> > A detailed description of the reasoning traces for the real-world experiments is provided in the Appendix (see "Extracting thoughts" paragraph for the real-world experiments). In summary, we followed the ClevrSkills format for providing short reasoning traces made of the current subtask and motion primitive. These are heuristically extraced using the proprioceptive information contained in each trajectory (e.g. when the gripper is closed is grasping an object).
> >
> > We did not optimize these traces specifically for the real-world settings. Instead, we hypothesized that our methodology and findings in ClevrSkills would transfer to the real-world, and we found this to hold empirically.

---

> ### Author Response · Authors · 2025-11-21
>
> > There appears to be substantial overlap with a contemporaneous, peer-reviewed publication (...). It would be great if the authors could clarify what is genuinely new here (theory, method, or empirical evidence) beyond this prior work.
>
> We acknowledge the importance of recognizing concurrent efforts in the field and we have updated our Related Work and Discussion to include a reference to the mentioned ECoT-Lite approach.
>
> We see two main similarities with the EcoT-Lite paper: (i) the findings are similar, i.e. learning from reasoning traces improve's the model performance, even when no reasoning traces are generated at test-time, (ii) the reasoning dropout method resembles an instance of HyT where only the "think" and "action" distributions are learned, with the difference that reasoning dropout can also drop only a part of the reasoning trace (similarly to what is done in the DualFormer paper [3]).
>
> We now discuss some of the major differences between ECoT-Lite and our work:
> * *Theory*: our approach presents an overarching framework for training VLAs that learn different conditional action probabilities. Our hypothesis originates from ideas in human cognition studies [4], and we present an approach with the clear intention of validating our hypothesis. In contrast, ECoT-Lite investigates several approaches that are related to ECoT (reasoning pre/co-training, dropout, reasoning scaffolding and thinking tokens), with the intention of better understanding which of these contributes to ECoT performance and deriving a training strategy that could match ECoT without predicting reasoning traces at inference time. The fact that reasoning dropout, which is the closest strategy to our approach, empirically performs best (along with reasoning pre-training) it's coincidental, and further supports our claims.
> * *Method*: our approach introduces a "follow" distribution, which is closely related to hierarchical approaches. This feature is unique to our work as hierarchical approaches are not discussed in the ECoT-Lite paper.
> * *Empirical evidence*: ECoT-Lite comparison focuses on testing different training strategies' performance against standard VLA and ECoT strategies, in simulation and real-world. Our work first investigates standard VLA, ECoT and hierarchical approaches (HiRobot), and instruction-following capabilities in the ClevrSkills benchmark. Then, it shows scalability to other VLA architectures and compares to state-of-the-art approaches in LIBERO. Finally, it demonstrates real-world applicability, providing a an extensive and thorough evaluation of our approach.
>
> ---
>
> We would like to thank the reviewer once again for contributing to the review process of our paper and we remain available for further clarifications.
>
> [1] Causal Confusion in Imitation Learning, De Haan et al, 2019
>
> [2] MolmoAct: Action Reasoning Models that can Reason in Space, Lee et al, 2025
>
> [3] Dualformer: Controllable Fast and Slow Thinking by Learning with Randomized Reasoning Traces, Su et al, 2024
>
> [4] Daniel Kahneman. Thinking, fast and slow. 2011.

---

> > ### Comment · Reviewer_cYRE · 2025-11-26
> >
> > I appreciate the authors' effort into addressing my concerns.
> >
> > Regarding weakness (1), the argument made by the authors in the rebuttal generally makes sense. If not done already, it would be great if the authors could update the main paper text to make the analysis about representation learning more clear.
> >
> > Regarding weakness (2), the authors' arguments about not needing to run additional experiments on these more reasoning heavy tasks due to them being out of the scope of their project makes sense.
> >
> > The remaining questions ((1)-(6)) have been adequately answered. I appreciate the authors' inclusion of new appendix sections to address questions (1) and (5).
> >
> > Overall my assessment of the paper remains roughly the same, and so I will keep my score.

---

### Author Response · Authors · 2025-12-03

We would like to thank the reviewers once again for reviewing our manuscript, for the generally positive assessment of our work, and for providing useful feedback to further improve our presentation.

Given the circumstances, it was not possible for all reviewers to engage in the discussion. We are glad that the reviewers who were able to provide a response to our comments (cYRE, PRFm) confirmed that our rebuttal adequately addressed their concerns. We would also like to emphasize that this originally led to an increased score (as stated in the comment of rev. PRFm).

In our rebuttal, we provided additional clarifications on the methodology and scope of the paper, included additional results to answer questions from the reviewers, and updated the manuscript to reference the works mentioned by the reviewers and to incorporate useful insights that emerged from the discussion.

As follows, we summarize some of the main concerns raised by the reviewers and how we have addressed them in our comments and updated manuscript. We refer to our method as HyT. We have updated the manuscript using blue text for additions and green text for edits.
* **How does HyT improve the model's understanding of the environment?** HyT learns multiple conditional action probabilities ("think", "follow", "act"). This combined signal enriches the model’s internal representation while also removing reliance on generated thoughts at inference time. This directly addresses the request to clarify how the method relates to representation learning (rev. cYRE W1) and helps explain why ECoT can underperform due to the compounding dependency on reasoning accuracy (rev. cYRE Q3). Additional comments on this matter have been added to the updated manuscript, as requested by rev. cYRE, in the Introduction and Discussion sections.
* **Ablation study of the HyT coefficients / learning components**: To answer this question, we introduced two additional ablation studies in the Appendix (Figures 6 and 7), where we vary the HyT coefficients used during training. These results indicate that the "think" component is most beneficial in low-data regimes, while including the "follow" component improves instruction adherence at larger scales. Balanced coefficient combinations such as the default (0.25, 0.5, 0.25) show strong and robust performance across data scales. These additions directly answer rev. PRFm’s W1/W3/Q2 and rev. cYRE’s Q1.
* **Insights into which tasks benefit from an internalized CoT the most**: HyT yields larger gains on harder, longer-horizon tasks and under low-data conditions. This was originally observed in the LIBERO results, where HyT improves performance mostly on the more complex LIBERO Goal and Long suites. In the updated manuscript’s Appendix, we provide additional detailed results on ClevrSkills, where tasks are divided into three complexity groups: easy, medium, and hard. These results highlight the importance of HyT for more complex tasks, where the performance gap with standard VLAs is larger compared to easier tasks. This directly addresses rev. PRFm’s question on task complexity and is consistent with rev. cYRE’s broader analysis request.
* **Clarifying that longer tasks could still benefit from explicit CoT**: As stated in the Discussion, we acknowledge that abstract, multi-step cognitive tasks (e.g., complex planning or tasks requiring mathematical reasoning) may still benefit from interleaving reasoning and actions at test time. However, these tasks go beyond our present scope. HyT shows that primitive tasks widely present in robotic manipulation—such as moving or stacking objects—can be solved robustly and with low inference costs by improving the model’s internal representation. To address the reviewers’ concerns (rev. cYRE W2; rev. LZtX W2), we have highlighted this in a more detailed paragraph of the Discussion.
* **Why does HiRobot+Oracle outperform HyT+Oracle in Figure 4?**
Compared to HyT, HiRobot (hierarchical approach) trains two different models: a high-level model and a low-level model. This design requires both models to be accurate in their predictions at test time. As a result, in standard settings, HiRobot’s performance is lower than HyT and ECoT (see Figure 3). However, when oracle instructions are provided, as in the instruction-following benchmark (Figure 4), the low-level model is more specialized than HyT in following instructions accurately. Learning the "follow" probability in HyT alleviates this issue, as shown in the newly presented ablations (see the updated manuscript’s Appendix), but the specialized model still performs best in these settings. This addresses rev. cYRE Q2 and rev. PRFm Q1.

---

> ### Author Response · Authors · 2025-12-03
>
> * **Costs of annotated reasoning data**: As originally stated in the Discussion, we agree that annotating data with embodied reasoning can be costly. However, we are convinced that this cost will become negligible as state-of-the-art vision and vision-language models become more and more performant on visual and spatial reasoning tasks. To respond to rev. LZtX W1, we have highlighted this in a more detailed paragraph of the Discussion.
> * **Comparison with concurrent work**: To respond to the reviewers’ requests (rev. cYRE Q6; rev. zK8v W2), we added references and mentions to ECoT-Lite, $\pi_{0.5}$, and OneTwoVLA. We recognize the importance of acknowledging related work. However, given the recency of these works, per ICLR policy (*), we refer to these efforts as concurrent.
>
> (*) ICLR policy on contemporaneous work states: papers published at peer-reviewed venues within two months of the full paper deadline (on/after July 24, 2025) need not be compared; arXiv-only papers are not required comparisons.

---

### Meta-Review · Area_Chair_vhfA · 2026-01-05

**Summary:**

I will list the most important comments that the reviewers noted during the review process:
1) It remains unclear what the specific reason is that reasoning trace generation can be avoided at test-time, yet the benefits of reasoning can still be obtained.
2) The authors missed trying their approach out on tasks involving some complex decision making/logical inference before picking.
3) There are no important ablation experiments: for the loss coefficients (wa:0.25, wτ:0.5, wf:0.25), for the Monte Carlo sampling approach, for the individual contributions of Lfollow and Lthink.
4) The "follow" mode may not be optimally implemented.
5) The paper lacks systematic investigation of which task characteristics benefit most from internalized CoT training versus standard VLA training.
6) No inference speed comparison with π0-FAST (or other baselines).
7) Whether HyT achieves comparable performance at lower latency or merely approximates ECoT under similar compute.

**Reviewer Concerns:**

The authors did some work during the rebuttal phase and addressed a significant part of the comments:
1) The analysis of the results: the authors give some additional explanations.
2) Ablation study: the authors added additional experiments and updated Appendix of the manuscript.
3) Follow mode: the authors provided an analysis of the new ablation study.
4) Analysis of internalized CoT: the authors provided detailed plots in the Appendix of the updated manuscript.
5) Performance: for ClevrSkills benchmark HyT has a 3x faster control cycle.

Two important points were not fully addressed:
1) Complex tasks: the authors didn’t provide additional experiments.
2) Inference speed: the authors didn’t address the weakness.
Nevertheless, the reviewers who expressed them were satisfied with the partial answers, and therefore I am inclined to accept this article.

**Reviewer Scores:**

1) Reviewer cYRE (score 6) explicitly confirmed that he would leave his initial score.
2) Reviewer PRFm (score 6) confirmed explicitly that he would raise his initial score.
3) Reviewer zK8v (score 4) could raise his score.
4) Reviewer LZtX (score 6) could raise his score.

---

### Decision · Program_Chairs · 2026-01-26

Accept (Poster)